# Two (or more) for one: Identifying classes of household energy- and water-saving measures to understand the potential for positive spillover

Angela Sanguinetti[ID]¹*, Claire McIlvennie², Marco Pritoni³, Susan Schneider⁴

1 Center for Water-Energy Efficiency, University of California, Davis, California, United States of America, 2 Rubenstein School of Environment and Natural Resources, University of Vermont, Burlington, Vermont, United States of America, 3 Lawrence Berkeley National Laboratory, Berkeley, California, United States of America, 4 Department of Psychology, University of the Pacific, Stockton, California, United States of America

* asanguinetti@ucdavis.edu

**Data Availability Statement:** All relevant data are within the paper and its Supporting information files.

## Abstract

A key component of behavior-based energy conservation programs is the identification of target behaviors. A common approach is to target behaviors with the greatest energy-saving potential. The concept of behavioral spillover introduces further considerations, namely that adoption of one energy-saving behavior may increase (or decrease) the likelihood of other energy-saving behaviors. This research aimed to identify and describe household energy- and water-saving measure classes within which positive spillover is likely to occur (e.g., adoption of energy-efficient appliances may correlate with adoption of water-efficient appliances), and explore demographic and psychographic predictors of each. Nearly 1,000 households in a California city were surveyed and asked to report whether they had adopted 75 different energy- and/or water-saving measures. Principal Component Analysis and Network Analysis based on correlations between adoption of these diverse measures revealed and characterized eight water-energy-saving measure classes: Water Conservation, Energy Conservation, Maintenance and Management, Efficient Appliance, Advanced Efficiency, Efficient Irrigation, Green Gardening, and Green Landscaping. Understanding these measure classes can help guide behavior-based energy program developers in selecting target behaviors and designing interventions.

## Introduction

Behavior change interventions aimed at the residential sector have been increasingly called upon to help reach sustainability goals. For example, research suggests household behavior changes, combined with energy-efficient technologies, could reduce total US residential energy consumption by up to 20% [1]. Attempting to capture this potential, behavioral programs such as home energy reports have become common [2].

**Funding:** This research was funded by the California Department of Water Resources (https://water.ca.gov/): Agreement Number 4600011065 (Principal Investigator Frank Loge, UC Davis Center for Water-Energy Efficiency) The funders had no role in study design, data collection and analysis, decision to publish, or preparation of the manuscript.

**Competing interests:** The authors have declared that no competing interests exist.

A key component of all behavioral interventions is the identification of target behaviors. Some research suggests interventions should target one or more specific behaviors [3, 4]. The question then becomes: Which ones? There could be many from which to choose, e.g., Boudet et al. [5] identified 261 household energy-saving behaviors. A common approach is to target behaviors with the greatest environmental impact [6–8]. Stern [9] also advises consideration of behavioral plasticity, which is the likelihood that individuals will adopt a given behavior.

The concept of behavioral spillover, defined as "the extent to which engaging in one behavior influences the probability of conducting a subsequent behavior" [10] (p. 574), introduces further considerations for target behavior selection. An intervention targeting one pro-environmental behavior may increase or decrease the likelihood of others (i.e., positive and negative spillover, respectively) [11, 12]. Depending on the magnitude of these effects, spillover could have significant implications for program design and evaluation. Interventions that trigger positive spillover could increase cost-effectiveness [13] and warrant increased investment [12]. Rather than prioritizing single, high impact behaviors, in some contexts it might be more fruitful to consider the total impact of classes of related behaviors within which positive spillover is likely to occur.

The first objective of this research was to identify classes of household water- and energy-saving measures within which positive spillover is likely to occur. In a survey of 976 California households, data were collected on engagement in 75 household energy- and water-saving measures. Analysis enabled classification based on frequently co-occurring measures. Temporal relationships between adoption of measures within a class were not considered but will be an important area for future research.

Further exploratory analyses identified potential "gateway measures", i.e., those particularly likely to lead to spillover within and between classes. Understanding potential gateway measures could help program designers nudge households toward adopting suites of conservation measures. Gateway measures might be low impact and thus overlooked in behavioral programs that target only high-impact measures. Nilsson et al. [10] argued: "If positive spillover can be reliably elicited, behaviors with small effects should not be ignored since they have the potential to influence other behaviors with more substantial effects on the environment" (p. 574).

Lastly, this research explored demographic and psychographic predictors of each identified measure class. Understanding distinctions between measure classes in terms of their potentially unique drivers and barriers can contribute to more effective and efficient interventions. For example, Steinhorst et al. [14] found that personal norms and self-efficacy completely mediated an observed spillover effect among pro-environmental behaviors. Layering more traditional market segmentation approaches with behavior segmentation (i.e., dividing behaviors into classes based on their relationships and characteristics) can support more tailored strategies. For example, Karlin et al. [15] identified different demographic and psychographic profiles for two household energy-saving measure classes: efficiency (most strongly predicted by homeownership) and curtailment (most strongly predicted by environmental values and energy bill consciousness).

## Understanding behavioral similarity

While the behavioral mechanisms responsible for spillover are still not well understood [10], research and theory generally suggest positive spillover is more likely to occur amongst "similar" behaviors [10–12, 16–18]. Behaviors can be similar in terms of various attributes, such as where and when they occur, resources required, and function. Attributes can be real or perceived, universal or idiosyncratic. A consistent understanding of what constitutes similar

behavior in the context of behavioral spillover (i.e., what types of similarity predict positive spillover) is lacking.

Margetts and Kashima [17] suggested that the resources required to perform behaviors may strongly determine behavioral similarity in the context of spillover, with spillover being more likely to occur between behaviors requiring similar resources (e.g., money as opposed to time or effort). Thøgersen and Olander [19] suggested that a common goal across multiple behaviors might be the most important factor involved in spillover. Truelove et al. [12] also seem to define the kind of behavioral similarity that leads to positive spillover as behaviors with a common goal.

The concepts of response generalization and response classes from the field of behavior analysis [20–22] may be useful in furthering understanding of behavioral similarity, and thus of spillover. A response class is a group of behaviors that have the same function (i.e., are functionally related to common antecedents and consequences). When one behavior in a response class is reinforced, the others also become more likely to occur in the future (this is the process of response generalization).

Thus, response generalization depends on an individual's history of reinforcement. Response classes differ across individuals to the extent that the social and instrumental consequences of those responses have differed in each person's experience. However, many consequences will be similar, especially within a shared culture. Thus, though response classes are idiosyncratic, there are likely to be general response classes that are common across individuals.

Truelove et al. [12] noted that those with more environmental knowledge might perceive similarity across more behaviors compared to those with less environmental knowledge. Thus, response generalization could occur across many diverse pro-environmental behaviors simply because they all share a function of protecting the environment. However, pro-environmental behaviors also have more immediate and personal consequences, compared to the indirect and long-term consequence of protecting the environment, and these will also influence the development of response classes. For example, curtailment of energy or water use in the home could mean sacrifices in preferred hygiene, comfort, or entertainment habits. More positively, it could bring financial savings.

## Classifying household conservation behaviors

Several approaches have been taken to classify household conservation behaviors into categories of similar measures that might also be considered response classes within which positive spillover is likely to occur. One approach is to define categories based on theoretically derived behavioral attributes; measures with similar attributes are grouped together (e.g., [5, 3, 23]). Another approach is to classify measures based on consumers' perceptions of behavioral similarity (e.g., [19, 24, 25]). A third approach, taken in the present research, is to distill classes of similar behaviors based on patterns in actual or reported behavior (e.g., [15, 19, 26]). The most systematic classifications and those covering larger sets of behaviors consider either household energy- or water-saving measures but not both.

Boudet et al. [5] classified 261 household energy-saving measures based on nine behavioral attributes that they hypothesized to be important differentiators based on social and behavioral theory: household function (e.g., thermal comfort, hygiene, entertainment), cost, energy savings, frequency, skill required, observability (visibility to others), locus of control (who can engage in the behavior), and home and appliance topography (where the behavior occurs and with what appliance). Using cluster analysis to group behaviors with similar attribute profiles, they identified four measure classes: family style, call an expert, household management, and weekend projects. No similar treatment has been given to household water-saving measures or

larger sets of pro-environmental behavior encompassing both energy- and water-saving measures.

Rather than using theoretically derived attributes to classify measures, Kneebone et al. [24] asked consumers to sort 44 water-savings actions into groups and explain their rationale. Multidimensional scaling analysis was used to identify three classes of similar behaviors based on how often they co-occurred in participants' groupings; these were: mostly indoor curtailment or habitual behaviors, outdoor garden and plant-related behaviors, and efficiency and maintenance behaviors. An additional eight subgroups of behaviors were identified, characterized by attributes such as behavior type, location, ease of participation, behavioral goal, and personal practices or preferences.

Karlin et al. [15] classified household energy-saving measures based on survey respondents' self-reported engagement in eight diverse measures. They used Principal Component Analysis to identify two factors that best explained the variance: curtailment (no cost, high frequency measures) and efficiency (low frequency investments and maintenance measures). Thøgersen and Olander [19] applied this method to a more diverse set of pro-environmental behaviors, including household energy- and water-saving measures as well as alternative (non-car) transportation, buying organic, and recycling, but with a relatively small set of behaviors spanning these multiple categories (17 total, including 1 water-saving measure and 4 energy-saving measures).

Classifications of large sets of energy- and water-saving measures are lacking in existing literature. Water and energy use often overlap in the home, thus a relatively high degree of spillover between the two, compared to less closely related domains (e.g., transportation behavior), would seem reasonable. In a recent study in Burbank, California, an intervention consisting of home water reports (HWR) with feedback on water consumption and tips about water conservation led to reductions in both water and electricity consumption, despite the fact that electricity-consuming behaviors were not targeted in the reports [13]. Only 26% of the electricity savings could be explained by water conservation activities (e.g., running only full loads in the dishwasher), which suggests there was spillover to non-water-related energy-saving measures.

The study reported in this paper was part of a follow-up to the study reported in Jessoe et al. [13]. It aimed to further explore the potential for positive behavioral spillover among household water and energy saving measures. This was accomplished through extensive survey research in conjunction with implementation of the WaterSmart, Inc. HWR report program in Riverside, California.

## Methodology

This section briefly reviews the HWR intervention, as background information, followed by a detailed description of the post-treatment survey which was the sole source of data for the present study, and then a detailed description of analyses. See [27] for a full description of the HWR intervention and analysis of water and energy consumption data.

### Smart water-energy savings project

The Center for Water-Energy Efficiency at University of California, Davis, partnered with WaterSmart Software, Inc. on a HWR project in two California cities. This project, called Smart Water-Energy Savings, aimed to quantify both water and energy savings associated with the HWR program. The current research focuses on just one of the cities, Riverside.

The HWR program ran from September 2016 to November 2017. Only single-family households with at least one year of observable water usage history at their current residence were eligible. Out of 56,000 eligible households, 14,359 were randomly assigned to HWR treatment, leaving 38,751 households as the control group. Treatment households were randomly

assigned to two groups: WaterSmart and Hot WaterSmart. The latter added a focus on hot water savings, which was hypothesized to lead to greater energy savings from natural gas.

The WaterSmart HWR program features customized reports delivered by mail or email, and an online portal where residents can learn more about their water use and ways to save. Each report included feedback about past water consumption and tips on how to conserve water in the future. WaterSmart Software, Inc. keeps a library of tips and determines which tips each household receives (e.g., if they know a household has a pool, they may give pool-related water-saving tips). The authors of this research were provided with the tip library but not information about which tips each household received.

## Post-treatment survey

The survey featured questions assessing self-reported engagement in 75 water- and/or energy-saving measures. These data were used to identify measure classes (across the whole sample including HWR treatment and control households), regardless of whether measures were adopted before or during the HWR program. The 75 measures assessed included many of the water-saving (including hot water-saving) measures promoted in the HWRs, as well as additional energy-saving measures identified in previous research, particularly [5].

To avoid overwhelming participants, questions used a checklist response option format and were presented in multiple sets based on household topography, using two prompt formats: one directed at actions (43 measures) and the other at investments (36 measures). For actions, four items read: *Which actions do you regularly take (1) at home*; (2) *while bathing/grooming*; (3) *in the kitchen*; (4) *in your yard* (if they had one)? For investments, two items read: *Which [energy-; water-] saving investments/measures do you have in* (1) *your home*; (2) *your yard*? Participants were instructed to mark all that applied and response option order was randomized except for a *None of the above* option, which was always displayed last.

The survey also collected information on participant demographics, housing characteristics, and psychographics. Three Likert-type items concerned general engagement with each water and energy savings: *I carefully examine my household water/energy bills; I have put a lot of effort into saving water/energy at home*; and *I wish I knew more about how to save water/energy at home*. The survey also inquired whether the participant was the household water and energy utility bill-payer. After responding to each of the action and investment items, respondents' reported measures reappeared with a prompt to consider their reasons for taking the measures and check all that apply; for actions: *Pressure from other member(s) of my household*; *To be efficient/save money*; *I feel guilty if I am wasteful*; *To care for the environment*, and for investments: *Someone else in my household made the decision/purchase*; *To be efficient/save money in the long term*; *I received a rebate*; *To care for the environment*. These items were inspired by hypotheses in [12] regarding relationships between initial measure adoption decision mode, causal attribution, and behavioral difficulty and the likelihood of subsequent spillover.

The survey was distributed via email when an email address was available, and otherwise by postal mail. Only one response per household was allowed. Each participant received a $20 Starbucks gift card. Out of 5,703 households recruited, 976 surveys were completed, a 17% response rate. After further data cleaning for analysis (described below), the final sample was 878. The survey instruments and methods were approved by the University of California-Davis Institutional Review Board, with a waiver of consent due to anonymized data collection and limited risks (Application #826774–5).

Table 1 describes the sample characteristics relative to Riverside County population. Females and homeowners are overrepresented. The overrepresentation of females is a common bias in survey research and the overrepresentation of owner-occupied housing is likely

**Table 1. Survey sample characteristics compared to Riverside County population; source: US census 2017 American Community Survey (ACS).**

|  | Sample | Population |
|---|---|---|
| **Gender** | 39% Male; 61% Female | 50% Female, 50% Male |
| **Age** | *M(SD)* = 50 (15) | *Med* = 45–54 |
| **Median Income** | $60–69,999 | $61,000 |
| **Median Education** | Associate's degree | Associate's degree |
| **Housing Tenure** | 83% Own | 65% Own |
| **Mean Household Size** | 3.5 | 3.28 [1] |

[1] 2015–2019 summary estimate.

due at least in part to the study inclusion criterion that there be one year of observable water usage history data for each household at their current residence.

## Analysis

Energy- and water-saving measure classes were identified using Principal Component Analysis (PCA). Household water- and energy-saving measures frequently selected by the same respondents loaded most strongly onto a common factor. It was hypothesized that some identified measure classes would include both water- and energy-saving measures, indicating how spillover from water- to energy-saving measures, or vice versa, may occur.

PCA is a statistical method to reduce complex datasets into fewer core components (i.e., factors) based on underlying patterns in the data. PCA has been used in prior spillover and behavior segmentation work [15, 18, 24]. Promax oblique rotation method was used, which allows for correlation between factors (as opposed to an orthogonal method than assumes uncorrelated factors); this enabled an analysis of the degree of correlation between resultant water-energy-saving measure classes (an indicator of spillover across measure classes).

The PCA was based on the correlation matrix of binary responses for the 75 energy- and water-saving measures (0 = not checked; 1 = checked) across the combined survey sample of control and HWR treatment households. Respondents were excluded if they did not have a yard, since that would influence the yard-related measures to load onto a common factor. Measures were assigned to a class based on their highest factor loading. Measures with no factor loadings above .32 were not assigned to a class (threshold suggested by [28]).

Resultant measure classes were defined and described in relation to each other in terms of common behavioral attributes. The nine attributes defined in [5] were considered (adapted to be inclusive of water-saving measures), as well as the concept of resources required ([17]; also adapted to include tools). Table 2 describes these attributes. We assessed which attributes helped define each measure class and which did not (i.e., where there was diversity among measures within a given class). These descriptions were formed inductively and qualitatively rather than using predefined attribute levels and coding.

Network analysis was used to visualize measure classes and help highlight potential for intra- and inter-class spillover. An undirected, weighted network of behaviors was created using MATLAB software. In the network, each classified measure was displayed as a node, color-coded by measure class per the PCA, and the size of the node was proportional to the frequency at which the behavior occurred in the sample. Links between nodes were used to represent significant correlations between pairs of behaviors (Pearson's correlation); two measures (nodes) were linked if their correlation was significant at the alpha = 0.05 level. The resulting network was graphed using the force-directed layout with an inverse weight effect, such that

**Table 2. Behavioral attributes of energy- and/or water-saving measures.**

| Attribute | Description |
|---|---|
| Resources Required | Objective, quantifiable resources (money, tools, effort/time) |
| Savings | Water and/or energy savings potential |
| Cost | Purchase price for investment measures |
| Frequency | How often the measure is likely to be performed |
| Skill Level | Amount of ability for an adult to perform (e.g., possible without reading instructions, skill with tools, need expert) |
| Observability | Degree to which others notice that the measure is performed |
| Locus of Decision | Household member(s) who can make the decision to adopt |
| Household Function | Service provided (e.g., comfort, hygiene, nourishment) |
| Home Topography | Where in the home or property it occurs |
| Appliance Topography | Relation to appliance category (e.g., large electric, water taps) |

*Source*: Adapted from adapted from [5] and [17].

links were weighted by the correlation between pairs of measures; the stronger the correlation, the shorter the link connecting them. To the authors' knowledge, this is the first application of network analysis to pro-environmental behavioral segmentation and spillover.

Measure classes were further described in terms of adopter characteristics, via hierarchical linear regression. A model was created for each measure class, where the dependent variable was the count of reported measures within the class for each participant. Four groups of variables were explored as predictors: demographics, housing characteristics, engagement, and measure motivations. Since different versions of motivation questions were used for action versus investment measures, only the relevant questions were used for each class (i.e., investment measure motivation questions for classes composed of only investment measures, action motivation questions for classes composed of only action measures, and both for mixed classes). The predictor variable groups were introduced one at a time in the model, starting with demographics. Significant predictors (alpha = .05) at each step were retained for all subsequent steps; variables not significant when first entered were left out of subsequent steps.

## Results and discussion

The PCA converged in nine iterations to reveal eight factors underlying self-reported participation in water-energy-saving measures. The criterion for factor selection was an Eigenvalue greater than 1.5. The value of 1.5 was selected after examining the Scree plot and because using an Eigenvalue criterion of 1 yielded many factors (24), Eigenvalue = 2 yielded few (3 factors).

Forty-five measures had a factor loading of at least .32 and thus were categorized as part of a measure class (per threshold given in [28]). Two measures (drip irrigation and reusing boiled water) loaded onto multiple classes (two each). This leaves 30 measures that did not load strongly enough onto a factor to be categorized in a measure class. This is disappointing from one angle, because some uncategorized measures (e.g., turn off computers when not using) seem similar to measures that did load highly on one of the eight factors (e.g., turn TV off when not using) and we do not know why. On the other hand, it narrows the focus down to measures with the most implications for spillover.

In support of the study hypothesis, several of the identified measure classes contain both energy- and water-saving measures (Maintenance & Management, Water Conservation, Energy Conservation and Edge of Efficiency). Table 3 shows the rotated component matrix,

**Table 3. PCA results: Factor loadings of each measure onto each measure class.**

| Water- and/or Energy-Saving Measure | Frequency (%) | Efficient Appliance | Maintenance & Management | Water Conservation | Efficient Irrigation | Green Landscaping | Green Gardening | Energy Conservation | Advanced Efficiency |
|---|---|---|---|---|---|---|---|---|---|
| ENERGY STAR TV | 63 | .82 | -.03 | .08 | -.04 | -.19 | .01 | -.02 | .09 |
| ENERGY STAR refrigerator | 68 | .76 | -.10 | .11 | -.02 | -.06 | .00 | -.01 | .05 |
| ENERGY STAR dryer | 63 | .76 | -.11 | .08 | -.04 | -.06 | -.04 | .06 | .06 |
| ENERGY STAR computer | 42 | .71 | -.02 | .07 | -.01 | -.03 | .02 | -.06 | .03 |
| Check for thermal leaks | 28 | -.18 | .72 | .06 | -.01 | -.16 | .05 | .03 | .10 |
| Caulk/seal doors/windows/baseboards | 36 | -.11 | .71 | -.02 | .03 | -.03 | .00 | .06 | -.03 |
| Check for shower/faucet/toilet leaks | 75 | .02 | .54 | .00 | .14 | -.09 | -.01 | .03 | -.20 |
| Weather-stripping on doors/windows | 43 | .18 | .51 | -.19 | -.06 | .16 | .05 | .00 | -.06 |
| Clean refrigerator coils | 29 | -.01 | .50 | .17 | .02 | -.06 | .02 | -.11 | .16 |
| Clean light bulbs | 28 | -.05 | .41 | .27 | -.03 | -.07 | -.08 | .04 | .14 |
| Low-flow faucet aerator(s) | 32 | .20 | .35 | .01 | -.07 | .29 | .02 | -.12 | -.08 |
| Set water heater temperature to 120°F | 39 | -.04 | .33 | .04 | .14 | -.08 | .06 | .15 | .15 |
| Turn off water while soaping hands | 36 | .05 | -.08 | .65 | -.01 | .14 | .01 | -.07 | .01 |
| "…" when scrubbing fruits and veg. | 51 | .11 | -.03 | .60 | .06 | .05 | .05 | .02 | -.08 |
| "…" while scrubbing face/hair/body | 38 | -.07 | -.02 | .54 | .05 | .17 | -.07 | .01 | .13 |
| "…" while scraping/scrubbing dishes | 68 | .04 | .04 | .54 | .02 | .03 | .01 | .07 | -.10 |
| "…" while shaving | 55 | .15 | .11 | .54 | .09 | .13 | -.12 | -.07 | -.05 |
| "…" while brushing teeth | 85 | .15 | -.05 | .45 | .18 | .04 | -.01 | .08 | -.26 |
| Take short showers (5 minutes or less) | 52 | .10 | .06 | .36 | -.08 | .03 | .04 | .06 | .15 |
| Reuse cooking water after boiling… | 21 | -.03 | .01 | .33 | .00 | .13 | .33 | -.08 | .02 |
| Check for irrig. system/sprinkler leaks | 64 | -.04 | .07 | .03 | .71 | -.01 | -.04 | .07 | -.06 |
| Trim plants around sprinkler heads | 59 | .00 | .13 | .02 | .68 | -.15 | .01 | .02 | -.08 |
| Rotating sprinkler heads | 30 | -.06 | .10 | .02 | .66 | -.05 | -.05 | -.24 | -.12 |
| Adjust irrig./sprinkler timer monthly | 42 | -.01 | .01 | .08 | .64 | -.09 | -.05 | -.04 | .14 |
| Multiple irrig./watering start times | 40 | -.05 | -.11 | -.04 | .59 | .15 | .04 | .10 | .17 |
| Weather-based irrigation controller | 9 | .06 | -.10 | .09 | .37 | .04 | -.07 | -.23 | .27 |
| Water only at dawn or dusk | 80 | .09 | -.13 | -.03 | .34 | -.13 | .14 | .29 | -.15 |
| Drip irrigation | 26 | -.06 | -.20 | -.06 | .32 | .45 | -.02 | .04 | .17 |
| Changed grass to native plants | 14 | -.12 | -.08 | .13 | -.20 | .77 | .12 | .08 | -.05 |
| Replaced high water use plants… | 30 | -.05 | -.12 | .16 | -.02 | .72 | .13 | .05 | -.15 |

*(Continued)*

**Table 3.** (Continued)

| Water- and/or Energy-Saving Measure | Frequency (%) | Efficient Appliance | Maintenance & Management | Water Conservation | Efficient Irrigation | Green Landscaping | Green Gardening | Energy Conservation | Advanced Efficiency |
|---|---|---|---|---|---|---|---|---|---|
| Replaced lawn with artificial turf | 3 | -.08 | .08 | .06 | .00 | .49 | -.33 | -.08 | -.08 |
| Mulch leaves and leave in yard. . . | 25 | -.02 | .08 | -.05 | -.15 | .02 | .71 | .02 | .09 |
| Compost grass/leaves/food. . . | 23 | -.03 | -.10 | .09 | -.05 | .08 | .66 | -.02 | .12 |
| Put mulch at base of tree/bush/shrub | 29 | .10 | -.05 | -.07 | .00 | .28 | .60 | -.02 | -.10 |
| Mulching lawnmower | 15 | .05 | .12 | -.10 | .07 | -.09 | .57 | -.14 | .03 |
| Water diff. plants according to needs | 63 | -.14 | .07 | .06 | .12 | .17 | .32 | .21 | -.09 |
| Turn AC down/off at night in summer | 79 | -.07 | -.12 | .07 | -.10 | .21 | .03 | .57 | .13 |
| Turn heat. down/off at night in winter | 75 | -.11 | .02 | -.02 | .08 | .08 | -.08 | .51 | .12 |
| Turn off TV when not in use | 94 | .01 | .08 | .18 | -.12 | .07 | -.18 | .39 | .06 |
| Fully load clothes washer | 86 | .07 | .07 | -.04 | -.08 | -.10 | .10 | .38 | -.05 |
| Reuse bath towels | 88 | -.06 | .10 | -.20 | .01 | .10 | .08 | .33 | -.21 |
| Tankless water heater | 6 | .11 | -.12 | -.03 | -.15 | -.11 | .03 | .10 | .63 |
| Hot water recirculation pump | 6 | -.01 | .04 | .04 | .10 | -.13 | .07 | -.15 | .54 |
| Water displacement device in toilet(s) | 11 | .04 | .11 | .05 | -.01 | -.09 | .12 | .01 | .44 |
| Smart thermostat | 26 | .14 | -.05 | -.06 | .00 | .22 | -.06 | .11 | .34 |
| High-efficiency showerhead | 48 | .31 | .22 | .07 | -.01 | .24 | -.01 | -.01 | -.08 |
| High-efficiency toilet | 46 | .29 | -.01 | .02 | -.05 | .30 | .02 | .12 | .13 |
| LED lights | 70 | .25 | -.03 | -.01 | .00 | .19 | .00 | .05 | .08 |
| Dryer with sensor | 41 | .21 | .05 | -.14 | .11 | .02 | .06 | .02 | .17 |
| Insulation around hot water tank | 29 | .13 | .30 | -.06 | -.01 | .21 | .11 | -.10 | .03 |
| Clean/replace A/C filters | 78 | .13 | .30 | -.06 | .13 | .03 | -.15 | .21 | -.06 |
| Insulation around hot water pipes | 23 | .10 | .24 | -.10 | .03 | .10 | -.07 | .02 | .31 |
| High-eff. or double-paned windows | 46 | .18 | .24 | -.24 | -.01 | .17 | .02 | .06 | -.06 |
| Water pressure regulator valves | 28 | .15 | .24 | .00 | .07 | .08 | .09 | -.10 | .15 |
| Insulation in walls, ceilings, roof, attic | 59 | .13 | .23 | -.08 | .07 | .17 | -.01 | .07 | .06 |
| Use broom instead of hose to clean driveways/walkways/decks/patios | 77 | -.04 | .22 | .24 | .00 | .06 | .02 | .22 | -.13 |
| Use cloth instead of hose to clean lawn furniture/outdoor toys/sports eq. | 44 | -.10 | .23 | .22 | -.04 | .00 | .03 | .21 | .11 |
| Capture cold water while wait. for hot | 10 | -.03 | .03 | .22 | -.01 | -.04 | .24 | -.03 | .21 |

(Continued)

**Table 3.** (Continued)

| Water- and/or Energy-Saving Measure | Frequency (%) | Efficient Appliance | Maintenance & Management | Water Conservation | Efficient Irrigation | Green Landscaping | Green Gardening | Energy Conservation | Advanced Efficiency |
|---|---|---|---|---|---|---|---|---|---|
| Stop watering when it rains | 89 | .11 | -.11 | .09 | .30 | -.24 | .17 | .22 | -.02 |
| Ensure water isn't running onto pave. | 70 | .04 | .06 | .15 | .23 | .06 | .09 | .30 | -.08 |
| Hose faucet timer | 9 | -.08 | .10 | -.06 | .23 | .00 | .15 | -.22 | .09 |
| Graywater system | 2 | -.11 | .03 | .19 | .02 | .30 | .00 | -.29 | -.02 |
| Permeable pavement | 5 | .00 | .02 | .15 | -.06 | .30 | .02 | -.10 | -.01 |
| Solar-powered garden lights | 26 | .03 | .04 | -.08 | .03 | .27 | .09 | .03 | -.08 |
| Rainwater catchment system | 5 | .03 | .04 | .15 | .11 | .20 | .16 | -.30 | -.08 |
| Soil moisture system | 1 | -.15 | .10 | .06 | .08 | .13 | -.04 | -.24 | .17 |
| Check soil moisture before watering | 28 | -.03 | .13 | .10 | .09 | -.13 | .28 | .14 | .20 |
| Turn off lights when leaving room | 95 | .08 | .03 | .22 | .02 | .02 | -.14 | .30 | -.07 |
| Close refrigerator door quickly | 89 | .05 | .16 | .14 | .02 | -.08 | -.09 | .29 | -.02 |
| Cover pots and pans when cooking | 81 | -.13 | .30 | .12 | -.11 | -.02 | .04 | .28 | -.04 |
| Fully load dishwasher (not all had) | 55 | .00 | -.03 | -.24 | .21 | .16 | -.09 | .22 | .20 |
| Turn off computers when not in use | 73 | .05 | .15 | .29 | -.09 | -.13 | -.08 | .20 | .09 |
| Motion sensor/dimmer/timer for lights | 32 | .01 | .06 | -.08 | .16 | .02 | .08 | -.04 | .31 |
| Whole house fan | 25 | .08 | .13 | .02 | -.05 | -.08 | .13 | .04 | .25 |
| Air dry laundry | 40 | -.10 | -.07 | .26 | -.05 | -.05 | .16 | .04 | .16 |

including all behaviors and their factor loadings (i.e., correlation with each identified measure class). Measures are sorted within each class in order of highest to lowest factor loadings. Measures with the highest factor loadings are most indicative of a class (i.e., overall most strongly correlated with other measures in the class). Thus, these are potential *intra*-class "gateway" measures that, when adopted, might be most likely to lead to positive spillover to other measures in the same class. For example, checking for thermal leaks was most representative of Maintenance & Management, and would likely be the highest leverage measure to target in an intervention promoting multiple Maintenance & Management measures. We hypothesize that the more common measures within a given class may precede the less common measures, but future research is needed to explore the temporal relationships between adoption of different measures within a measure class.

Fig 1 illustrates the eight measure classes. Table 4 presents an overview of how the behavioral attributes used by Boudet et al. [5] to categorize household energy-saving measures are useful in defining the measure classes identified in the PCA. Checked cells indicate a common attribute and empty cells indicate diversity within the measure class. Some classes are homogenous in terms of many attributes, while others are characterized by fewer common attributes. For example, Efficient Appliance measures require a common resource (money); have relatively high potential savings; are relatively expensive, infrequent, low skill, and observable; and are generally available only to adult household members. On the other hand, Edge of

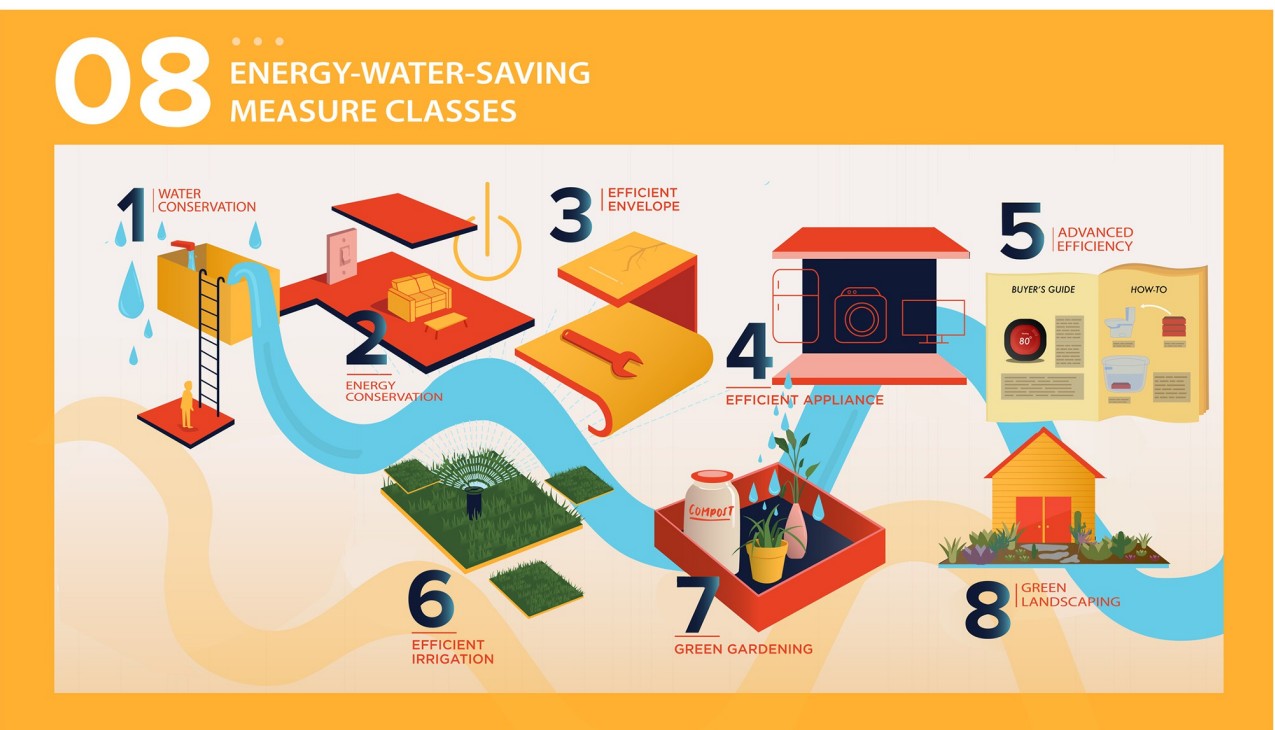

**Fig 1. Illustration of identified measure classes.**

Efficiency measures (smart thermostat, tankless water heater, hot water recirculation pump and toilet tank water displacement device) are infrequent measures taken by adults only, and beyond that they have little in common. This class seems to showcase appliances at the next level of innovation in energy or water efficiency, as well as more obscure add-on measures. This might be indicative of a special type of required resource: knowledge of the existence of the measures. This is speculation that should be explored in future research.

Table 5 shows the correlations between measure classes, which has implications for *inter*-class spillover. Specifically, spillover might be more likely between highly correlated measure

**Table 4. Distinguishing behavioral attributes (per Boudet et al., 2016) of identified water-energy-saving measure classes.**

| | Resources required | Energy and/or water savings | Cost | Occurrence frequency | Skill | Observability | Locus of decision | Household function | Home topography | Appliance topography |
|---|---|---|---|---|---|---|---|---|---|---|
| Advanced Efficiency | | | | X | | | X | | | |
| Efficient Appliance | X | X | X | X | X | X | X | | | |
| Maintenance & Management | X | | X | | | X | X | | | |
| Energy Conservation | X | X | X | X | X | X | X | | | |
| Water Conservation | X | X | X | X | X | X | X | X | X | X |
| Efficient Irrigation | | | | | | | X | X | X | X |
| Green Gardening | | | | | | | X | X | X | X |
| Green Landscaping | X | X | X | X | X | X | X | X | X | X |

**Table 5. Correlations among water-energy-saving measure classes.**

| | Advanced Efficiency | Efficient Appliance | Maintenance & Management | Energy Conservation | Water Conservation | Efficient Irrigation | Green Gardening | Green Landscaping |
|---|---|---|---|---|---|---|---|---|
| Advanced Efficiency | 1 | 0.21 | 0.30 | 0.10 | -0.03 | 0.28 | 0.11 | 0.36 |
| Efficient Appliance | | 1 | 0.44 | 0.28 | -0.12 | 0.39 | 0.17 | 0.38 |
| Maintenance & Management | | | 1 | 0.25 | 0.11 | 0.36 | 0.26 | 0.36 |
| Energy Conservation | | | | 1 | 0.03 | 0.31 | 0.16 | 0.16 |
| Water Conservation | | | | | 1 | -0.06 | 0.15 | -0.18 |
| Efficient Irrigation | | | | | | 1 | 0.23 | 0.37 |
| Green Gardening | | | | | | | 1 | 0.21 |
| Green Landscape | | | | | | | | 1 |

classes and less likely between classes with smaller correlations. The highest correlation is between Efficient Appliance and Maintenance & Management, suggesting that interventions targeting one of these classes should consider positive spillover to measures in the other class.

Fig 2 shows the network analysis of all 75 measures. Intra-class links, representing significant correlations between measure pairs within the same class, are color-coded by measure class. Inter-class links, representing significant correlations between measures in different classes, are light gray. Spatial positioning of classes in relation to each other and the spread of measures within each class is indicative of the potential for intra- and inter-class spillover. A class densely clustered away from other classes, like Water Conservation, suggests high potential for intra-class spillover and low potential for inter-class spillover. A high degree of overlap between classes, like Efficient Appliance, Efficient Irrigation, and Maintenance & Management, suggests potential for interclass spillover.

## Measure class adopter profiles

Table 6 provides descriptive statistics for the independent variables entered in the regression models in Steps 3 and 4 (Table 1 summarized the variables in Steps 1 and 2). Table 7 presents

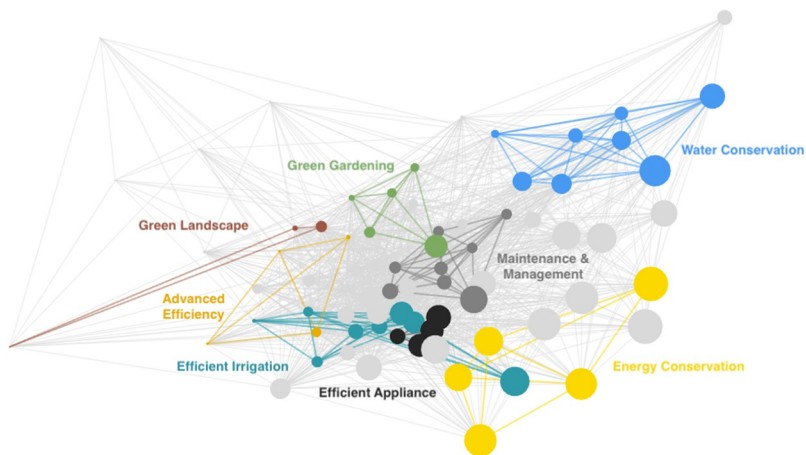

**Fig 2. Network analysis depicting correlations among self-reported household energy- and water-saving measures.**

**Table 6. Survey sample general engagement and motivations with respect to household water and energy savings.**

| Independent Variable | Frequency or Median |
|---|---|
| **Step 3: General Engagement** | |
| Bill-payer | 82% |
| Carefully examines energy bills | Strongly agree |
| Puts effort in saving energy | Somewhat agree |
| Wants to know how to save energy | Somewhat agree |
| Carefully examines water bills | Strongly agree |
| Puts effort in saving water | Somewhat agree |
| Wants to know how to save water | Somewhat agree |
| **Step 4: Measure Motivations** | |
| *Action Measure Motives* | |
| Pressure from other member(s) of my household | 6% |
| To be efficient/save money | 92% |
| To care for the environment | 71% |
| Feels guilty if wasteful | 48% |
| *Investment Measure Motives* | |
| Someone else in my household made the decision/purchase | 12% |
| To be efficient/save money | 87% |
| To care for the environment | 62% |
| Received rebate | 15% |

the final regression model for each identified measure class. Age was the most common demographic predictor of measure class scores, always revealing a positive relationship between age and engagement in measure classes. Higher income predicted four measure classes, while lower income predicted Water Conservation. Homeownership also predicted four measure classes, particularly those that include higher cost investment measures, consistent with [15].

In terms of general engagement with household energy and water use, being the bill-payer was only a significant predictor of greater engagement in Energy Conservation measures, and careful attention to energy and water bills did not predict adoption of any measure class. This is consistent with the hypothesis that a calculation-based decision mode (based on consideration of costs and benefits) for measure adoption is not expected to consistently lead to spillover [12]. Reporting effortful engagement in saving energy or water was more often predictive of adoption of measure classes. This relates to the concept of behavioral difficulty. Truelove et al. hypothesized that if an initial behavior requires substantial effort, it is more likely to affect an adopter's self-identity and spill over to additional behaviors perceived as consistent with that identity. Thus, perceptions of effort might predict adoption of any measure class, as each represents a case of positive spillover. Other types of effort not accounted for in this study include financial investment.

In terms of motivations for measure adoption, caring for the environment was predictive of every measure class, whereas guilt, social pressure and rebates were much less so. These findings are consistent with the hypotheses of Truelove et al. [12] regarding relationships between decision modes and causal attributions for adopting an initial behavior and the likelihood of subsequent spillover. In particular, they suggested that spillover is more likely to occur when the decision to adopt an initial measure is based on a rule or role (e.g., being an environmentalist) or attributed to related internal motivations, rather than affective decisions (e.g., based on guilt) and external causes (e.g., social pressure or price signals). Future research into adopter profiles should consider additional demographic, psychographic, and contextual variables to deepen understanding of these measure classes.

**Table 7. Regression models exploring predictors of measure class adoption.**

| | Advanced Efficiency | Efficient Appliance | Maintenance & Management | Energy Conservation | Water Conservation | Efficient Irrigation | Green Gardening | Green Landscape |
|---|---|---|---|---|---|---|---|---|
| Intercept | | 1.1*** | 2.0*** | 2.8*** | 6.3*** | 1.5** | 1.3*** | |
| *Step 1: Demographics* | | | | | | | | |
| Gender (1 = Male) | .11* | .22* | | | | | | |
| Age | .00** | .01** | .03*** | .01*** | | .02*** | .01* | .00* |
| Income | .04*** | .09*** | | | -.06** | .17*** | | .02** |
| Education | | | -.16*** | .07*** | -.15** | | .05 | |
| *Step 2: Housing Characteristics* | | | | | | | | |
| Tenure (1 = Own) | .16* | .28* | | | | .39* | .43** | |
| Household size | | | | | | -.07 | | |
| *Step 3: General Engagement* | | | | | | | | |
| Bill-payer | | | .33 | .34*** | | | | |
| Carefully examines energy bills | | | | | | | | |
| Puts effort in saving energy | | | .51*** | | .27* | | | |
| Wants to know how to save energy | | | -.17* | | | | | |
| Carefully examines water bills | | | | | | | | |
| Puts effort in saving water | | | | | .39*** | | | |
| Wants to know how to save water | .04 | | | | .14* | | | |
| *Step 4: Measure Motivations* | | | | | | | | |
| Social pressure[a, b, or a+b] | | | | | | | | |
| To be efficient/save money[c, d, or c+d] | .18* | .39* | .27* | .33** | -.51* | | | |
| To care for environment[e, f, or e+f] | .18*** | .33** | .25** | .20** | .80*** | .21* | .27*** | .21*** |
| *Action Measure Motive* | | | | | | | | |
| Feels guilty if wasteful | | | | .15* | | | | |
| *Investment Measure Motive* | | | | | | | | |
| Received rebate | | | | | | .40* | | .17* |
| ***Model R²*** | **.087** | **.117** | **.157** | **.090** | **.173** | **.160** | **.060** | **.043** |

[a] Pressure from other member(s) of my household (action measures).

[b] Someone else in my household made the decision/purchase (investment measures).

[c] To be efficient/save money (action measures).

[d] To be efficient/save money (investment measures).

[e] To care for the environment (action measures).

[f] To care for the environment (investment measures).

## Limitations

Spillover has a temporal dimension (e.g., one behavior leading to another) that was not addressed in this study. The PCA focused on identifying categories of often-co-occurring measures. We cannot say the order in which measures were adopted, but we can say that if someone engages in one measure within an identified class, they are more likely to also engage in the other measures in that class. Our assertion that positive spillover is likely to occur within response classes is consistent with the definitions of both concepts and observations by other behavioral spillover researchers, e.g.,: "The existence of such behavioural categories may in itself be taken as an indication that some transfer of environment-friendly conduct goes on between behaviours that are closely associated"([19], p. 234).

Another limitation was an exclusive focus on positive spillover. Only one negative factor loading (-0.33 for "replaced lawn with artificial turf" on Green Gardening) exceeded the magnitude required for positive factor loadings for a measure to be included in a class. This makes sense because Green Gardening included measures that involve lawns (e.g., mulching lawnmower). If this level of negative correlation is indicative of negative spillover, the results suggest there is little risk of negative spillover amongst the assessed measures. The largest magnitude negative interclass correlation is -0.18 between Water Conservation and Green Landscape, which is generally considered a weak correlation, but reaches statistical significance ($t(876) = -5.42$, $p < .0001$). It is possible that there is some moral licensing or single action bias whereby households that invest in Green Landscape measures are less likely to engage in Water Conservation measures, but further research should use methods directly aimed at assessing negative spillover.

The generalizability of findings may be somewhat limited due to the specific geographical context. There could be geographically based differences in terms of the measures within a given class because different measures may be available in different places. However, the measure classes themselves should be less affected by these variations because adoption of whichever measures within a given class that are available should correlate, the exception being if there are multiple unique and correlated measures that could form an additional measure class.

This study did not assess all energy-water-saving measures identified in prior research (e.g., [5]). For example, we did not include ENERGY STAR dishwasher or clothes washer, which might have loaded with Efficient Appliance, proving it to be another category that includes both energy- and water-saving measures. Future research that includes additional measures is needed to confirm and possibly expand upon the eight measure classes identified in this study.

Finally, surveying households in the context of the HWR program may have influenced the results. Relying on self-reported behaviors could have introduced response error, and demand characteristics are a particular concern among treatment participants who may have over-reported engaging in measures that were promoted in the HWRs. It is important to note that the absolute and relative frequencies of conservation measures reported by this sample may not be generalizable. A more specific issue concerns two measures that were offered for free in the HWR program (opt-in): low-flow faucet aerators and a high-efficiency showerhead. Many households who reported these measures adopted them within the program period (per a follow-up question asked for each investment measure). For those mainly motivated by that price signal, spillover might be less likely. This may be why high-efficiency showerhead was correlated with Efficient Appliance but under the .32 factor loading threshold to be included in that class.

To test whether and how the HWR treatment may have influenced the PCA results, we performed a separate PCA on only data from control group members, which comprised a

relatively small subset of the sample ($n$ = 163), to compare to the overall model dominated by data from treatment participants. The results were not easily interpretable, which is likely due to the small sample size [29]. Replications of this research in other contexts are needed to validate the results.

## Conclusion

People tend to concentrate their household energy- and/or water-saving efforts within some measure classes and not others. This research builds on prior energy behavior segmentation and spillover research by classifying 75 energy- and/or water-saving measures into 8 classes of similar measures within which positive spillover may occur. Past important work in this area has included deductive classifications of large sets of energy measures [5] and inductive classifications of relatively limited sets of measures (e.g., [15]). For example, with a larger set of measures, this research was able to validate Maintenance & Management as a distinct class, which was hypothesized but not supported in Karlin et al. [15]. We also provided a more differentiated classification that complements the four categories defined by Boudet et al. [5] (family style, call an expert, household management, and weekend projects). For example, our classes of Water Conservation and Energy Conservation correspond to Family Style.

Our classification confirms the importance of previously defined behavioral attributes (e.g., frequency, skill, cost) in determining the kinds of behavioral similarity that underlie spillover and highlights how different attributes are more, or less, useful in defining different categories. The weighting of various attributes in determining these response classes cannot be predetermined. Thus, more inductive research is required to continue to build our understanding of pro-environmental response classes. The novel application of network analysis in this research proved a useful visualization tool and should be integrated into future research on pro-environmental response classes and behavioral spillover.

Understanding these measure classes can inform behavioral programs targeting household energy and water conservation. Programs could systematically target measure classes, e.g., a series of energy reports focusing on one measure class at a time, each report promoting multiple measures within a given class and highlighting "gateway" measures. Programs with a central target measure could be leveraged to also promote related measures that consumers would be likely to adopt if they adopt the target measure. If program designers collect baseline data, they could identify measures that the target audience (individually or collectively) might be more inclined to adopt (i.e., from classes within which they have already adopted some but not all measures). Energy and water utility companies could partner with product manufacturers to bundle related appliances and devices and provide a rebate for the set. Overall, understanding pro-environmental behavior measure classes can enable strategic selection of target behaviors and support more tailored and cost-effective programs.

## Supporting information

**S1 File. Treatment group survey.**
(PDF)

**S2 File. Variable answer key.**
(DOCX)

**S1 Data. Smart water energy survey data.**
(XLSX)

## Author Contributions

**Conceptualization:** Angela Sanguinetti, Susan Schneider.

**Formal analysis:** Angela Sanguinetti, Claire McIlvennie, Marco Pritoni.

**Investigation:** Angela Sanguinetti.

**Methodology:** Angela Sanguinetti, Claire McIlvennie, Marco Pritoni.

**Software:** Marco Pritoni.

**Supervision:** Angela Sanguinetti, Marco Pritoni.

**Visualization:** Angela Sanguinetti, Claire McIlvennie.

**Writing – original draft:** Angela Sanguinetti, Claire McIlvennie.

**Writing – review & editing:** Angela Sanguinetti, Susan Schneider.

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
