## [Decision Letter · Decision Letter 0]

21 Apr 2021

PONE-D-21-05773

Two (or More) for One: Identifying Classes of Household Energy- and Water-Saving Measures to Understand the Potential for Positive Spillover

PLOS ONE

Dear Dr. Sanguinetti,

Thank you for submitting your manuscript to PLOS ONE. After careful consideration, we feel that it has merit but does not fully meet PLOS ONE’s publication criteria as it currently stands. Therefore, we invite you to submit a revised version of the manuscript that addresses the points raised during the review process.

Both reviewers provided what appear to be thoughtful and reasonable comments that, if fully addressed, should improve the manuscript. Please fully address each one. 

We look forward to receiving your revised manuscript.

Kind regards,

Damian Adams

Academic Editor

PLOS ONE

Additional Editor Comments:

Thank you very much for your revised submission. Based on my reading of the reviewer comments, I believe that a major revisions decision is appropriate. Both reviewers provided informative and reasonable comments. Most of the reviewer comments are minor in nature, however there are a couple that relate to the modeling and may require additional analysis. There are also several suggestions for improving the readability of the paper more generally. As you prepare your revision, please be sure to carefully address each of the reviewer comments, and to clearly identify in your response document how each was handled. Thank you very much,

Damian Adams

Journal Requirements:

4. We noted in your submission details that a portion of your manuscript may have been presented or published elsewhere.

"We published some of the basic results in a conference paper, but this paper includes major novel data analysis, including network analysis and regression analyses."

5. Please amend your manuscript to include your abstract after the title page.

6. Please ensure that you refer to Figure 2 in your text as, if accepted, production will need this reference to link the reader to the figure.

Reviewers' comments:

Reviewer's Responses to Questions

**Comments to the Author**

1. Is the manuscript technically sound, and do the data support the conclusions?

Reviewer #1: Yes

Reviewer #2: Yes

2. Has the statistical analysis been performed appropriately and rigorously? 

Reviewer #1: Yes

Reviewer #2: Yes

3. Have the authors made all data underlying the findings in their manuscript fully available?

Reviewer #1: Yes

Reviewer #2: Yes

4. Is the manuscript presented in an intelligible fashion and written in standard English?

Reviewer #1: Yes

Reviewer #2: Yes

5. Review Comments to the Author

Reviewer #1: The goal of this manuscript is to identify energy- and water-saving measure classes within which positive behavioral spillovers may occur. Using cluster analysis to analyze survey data from 1,000 households in a city in California that reported adoption of 75 energy- and/or water- saving measures, the authors identified eight water-energy-saving measure classes. The manuscript is generally well written and the methods used are appropriate. Some specific comments and suggestions for the authors are provided below:

1. The authors should state the goal of the paper much earlier in the introduction. I was on page three of the introduction and still found myself wondering what this paper does or how it contributes to the literature described in the introduction.

2. Which city was the HWR program implemented? How big is the city?

3. In your literature review you report studies that explored either energy-saving measure classes or water-saving measure classes. Have prior studies explored both water-energy-saving measure classes?

4. Your sample characteristics are reported compared to population of California or population of the US?

5. What are the summary statistics of other behaviors included in the analysis in regression analysis in Table 6?

6. Abbreviations under Table 1 should be spelled out.

7. How generalizable are your findings beyond a city in Riverside County, California?

8. How does the fact that the behaviors are self-reported impact your results? Could this introduce any bias in your analysis?

9. What is the impact of participant in the program on the survey responses? In other words, do you expect that participants tend to over- or under- estimate certain water- and energy- saving behaviors? How would this impact your results?

10. The use of network analysis is a great way to visualize measure classes and help highlight potential for intra- and inter-class spillovers. The fact that this is the first application of network analysis in this context should be highlighted as a contribution in the conclusion in addition to mentioning it on p. 13.

11. The fact that assignment into treatment and control groups within HWR program could have influenced your results is a serious concern. Have you done the analysis separately on treatment and control groups? How are they different?

12. p. 7: change Jessoe and colleagues (2017) to Jessoe et al. (2017)

Reviewer #2: Two (or more) for one: identifying classes of household energy-and water-saving measures to understand the potential for positive spillover

This study analyzed data from survey of 1,000 California residents and grouped into eight different classes using cluster analysis. Such classification is helpful to targeting specific groups and designing specific intervention based on their characteristics, therefore, tailoring cost-effective programs. The study has uniquely used a network analysis technique to visualize measure classes identified in multivariate analysis. The authors thoroughly discuss the results and provide important management implications. Here are my suggestions:

• Table 1 clearly shows the difference in population and sample in terms of % of female, income, education, housing tenure, household size, did you do non-response bias check? You should mention why the sample is quite different than population.

• Figure 1 can be deleted as it is already explained in the text.

• In page 18, first para, should not this be “figure 2” as you are referring to eight classes in the figure?

• Consider revising the Table 4 as it included the categories used by Boudet et al. When I look at the title, it seems all the table items are from this study.

• In-text reference is not consistent. For instance, page 6, para 1 (comma separate two citations)

• Italicized the survey questions to make it distinct than other text in the manuscript?

• My suggestion is to provide IRB information or at least mentioned that survey protocol was approved by IRB

• Different Likert scales were used for different items in page 10. Authors should clearly mention the Likert scale type or levels used in the survey.

• Page 11, authors said that items with factor loading more than 0.32 were included in the analysis. But, a common practice is considering factor loading over 0.40 (e.g.Vaske, 2008).

• The reference list has to be revised thoroughly by following the journal guideline. Citations in reference list are also not consistent. For example, Consortium for Energy Efficiency. (2018 mentioned “posted”, Frankel et al. 2013 menionted “retrieved”; Cooper et al. 2007, Hawken 2017 is not complete; DOI for several references is missing

• It should be helpful for readers if a copy of survey is available as supplement material

• It should be be ….validated in Karlin et al. (2014) in page 27?

• Consider changing title of Fig. 3 as it does not provide detail information.

6. PLOS authors have the option to publish the peer review history of their article (what does this mean?). If published, this will include your full peer review and any attached files.

Reviewer #1: No

Reviewer #2: No

---

## [Author Response · Author response to Decision Letter 0]

2 Nov 2021

Response to Reviewers

Dear editor and reviewers, 

We are grateful for your thoughtful feedback, suggestions, and guidance. We have addressed each comment carefully and hopefully to your satisfaction. We certainly feel it is a much stronger paper now as a result.

Editor/journal feedback

1. Format in journal style; templates:

Done.

Done.

We have included the survey instrument for the treatment group as supporting material. This version is similar to the Control group version except that it also includes a few questions about the home water report program.

Included.

4. We noted in your submission details that a portion of your manuscript may have been presented or published elsewhere.

"We published some of the basic results in a conference paper, but this paper includes major novel data analysis, including network analysis and regression analyses."

The following explanation has been added to the portal and cover letter: The former output was peer-reviewed and is included in the conference proceedings for the American Council for an Energy Efficient Economy (ACEEE) 2020 convention, which is not formally published; the papers are given to each attendee on a CD and published on the organization’s website, but not assigned a DOI or indexed in any academic database. The only overlap with this paper is the Principal Component Analysis of energy- and water-saving measures. The conference paper does not include the network analysis or the regression analysis. It was necessary to present the PCA again in this paper because it is the basis of the network analysis and regression analysis. As Reviewer 1 notes, the network analysis is a very novel approach worth highlighting. The regression analysis also adds considerably to our understanding of the measure classes.

5. Please amend your manuscript to include your abstract after the title page.

Added.

6. Please ensure that you refer to Figure 2 in your text as, if accepted, production will need this reference to link the reader to the figure.

Done.

Reviewer #1

1. The authors should state the goal of the paper much earlier in the introduction. I was on page three of the introduction and still found myself wondering what this paper does or how it contributes to the literature described in the introduction.

We extensively revised the introduction section in order to describe the research aims much earlier.

2. Which city was the HWR program implemented? How big is the city?

City of Riverside, population ~330,000. This information has been added.

3. In your literature review you report studies that explored either energy-saving measure classes or water-saving measure classes. Have prior studies explored both water-energy-saving measure classes?

Very few and none that are as remarkable as the ones described, and thank you for highlighting that this point was unclear. We have extensively revised this section of the literature review to make that clear.

4. Your sample characteristics are reported compared to population of California or population of the US?

It was a mix defined by table notes, but we have updated to reflect Riverside County to be a more precise estimate of sample representativeness relative to the+ treatment-eligible population.

5. What are the summary statistics of other behaviors included in the analysis in regression analysis in Table 6?

We have added these in a Table (now Table 6).

6. Abbreviations under Table 1 should be spelled out.

Addressed.

7. How generalizable are your findings beyond a city in Riverside County, California?

We now address this in the limitations section: The generalizability of findings may be somewhat limited due to the specific geographical context. There could be geographically based differences in terms of the measures within a given class because different measures may be available in different places. However, the measure classes themselves should be less affected by these variations because adoption of whichever measures within a given class that are available should correlate. 

8. How does the fact that the behaviors are self-reported impact your results? Could this introduce any bias in your analysis? 9. What is the impact of participant in the program on the survey responses? In other words, do you expect that participants tend to over- or under- estimate certain water- and energy- saving behaviors? How would this impact your results?

We now address these issues more fully in the limitations section (our response to question 11 is also relevant here): ...surveying households in the context of the HWR program may have influenced the results. Relying on self-reported behaviors could have introduced response error, and demand characteristics are a particular concern among treatment participants who may have over-reported engaging in measures that were promoted in the HWRs. 

10. The use of network analysis is a great way to visualize measure classes and help highlight potential for intra- and inter-class spillovers. The fact that this is the first application of network analysis in this context should be highlighted as a contribution in the conclusion in addition to mentioning it on p. 13.

Done.

11. The fact that assignment into treatment and control groups within HWR program could have influenced your results is a serious concern. Have you done the analysis separately on treatment and control groups? How are they different?

This is an important point. We did do a separate analysis but unfortunately we think the sample size of the control group was too small to yield a meaningful result. We expanded this discussion in the limitations section: To test whether and how the HWR treatment may have influenced the PCA results, we performed a separate PCA on only data from control group members, which comprised a relatively small subset of the sample (n = 163), to compare to the overall model dominated by data from treatment participants. The results were not easily interpretable, which is likely due to the small sample size [29]. Replications of this research in other contexts are needed to validate the results.

12. p. 7: change Jessoe and colleagues (2017) to Jessoe et al. (2017)

Done and changed the two other occurrences of “and colleagues” in other citations as well.

Reviewer #2

1. Table 1 clearly shows the difference in population and sample in terms of % of female, income, education, housing tenure, household size, did you do non-response bias check? You should mention why the sample is quite different than population.

We did not do a non-response bias check; however, we have edited the population statistics in the table to much more closely reflect the local population (of treatment eligible households). Now only gender and housing tenure are markedly different and we speculate the reasons for this when the table is introduced.

2. Figure 1 can be deleted as it is already explained in the text.

Done.

3. In page 18, first para, should not this be “figure 2” as you are referring to eight classes in the figure?

Yes, thanks for catching that typo, although now it is Figure 1 again given the deletion of the original Figure 1.

4. Consider revising the Table 4 as it included the categories used by Boudet et al. When I look at the title, it seems all the table items are from this study.

Table title has been revised to be more clear.

5. In-text reference is not consistent. For instance, page 6, para 1 (comma separate two citations)

Citations have been re-formatted per journal style guidelines.

6. Italicized the survey questions to make it distinct than other text in the manuscript?

Done.

7. My suggestion is to provide IRB information or at least mentioned that survey protocol was approved by IRB

Done.

8. Different Likert scales were used for different items in page 10. Authors should clearly mention the Likert scale type or levels used in the survey.

We now include the survey instrument as supporting material.

9. Page 11, authors said that items with factor loading more than 0.32 were included in the analysis. But, a common practice is considering factor loading over 0.40 (e.g.Vaske, 2008).

We provide a reference to support our use of 0.32. 

10. The reference list has to be revised thoroughly by following the journal guideline. Citations in reference list are also not consistent. For example, Consortium for Energy Efficiency. (2018 mentioned “posted”, Frankel et al. 2013 menionted “retrieved”; Cooper et al. 2007, Hawken 2017 is not complete; DOI for several references is missing

Citations and references have been re-formatted per journal style guidelines.

11. It should be helpful for readers if a copy of survey is available as supplement material

Done.

12. It should be be ….validated in Karlin et al. (2014) in page 27?

No, Karlin et al. hypothesized that there would be three categories of household energy-saving measures: efficiency, curtailment, and maintenance, but they only found curtailment and efficiency. Measures they thought would load onto a third “maintenance” factor loaded with efficiency measures. We think we identified the category because we had so many more measures in our analysis (75, compared to their 8).

13. Consider changing title of Fig. 3 as it does not provide detail information.

Good point. It is has been revised to provide more detail.

---

## [Decision Letter · Decision Letter 1]

11 May 2022

Two (or More) for One: Identifying Classes of Household Energy- and Water-Saving Measures to Understand the Potential for Positive Spillover

PONE-D-21-05773R1

Dear Dr. Sanguinetti,

We’re pleased to inform you that your manuscript has been judged scientifically suitable for publication and will be formally accepted for publication once it meets all outstanding technical requirements.

Kind regards,

Vassilis G. Aschonitis

Academic Editor

PLOS ONE

Additional Editor Comments (optional):

Reviewers' comments:

Reviewer's Responses to Questions

**Comments to the Author**

1. If the authors have adequately addressed your comments raised in a previous round of review and you feel that this manuscript is now acceptable for publication, you may indicate that here to bypass the “Comments to the Author” section, enter your conflict of interest statement in the “Confidential to Editor” section, and submit your "Accept" recommendation.

Reviewer #1: All comments have been addressed

Reviewer #2: All comments have been addressed

2. Is the manuscript technically sound, and do the data support the conclusions?

Reviewer #1: Yes

Reviewer #2: Yes

3. Has the statistical analysis been performed appropriately and rigorously? 

Reviewer #1: Yes

Reviewer #2: Yes

4. Have the authors made all data underlying the findings in their manuscript fully available?

Reviewer #1: Yes

Reviewer #2: Yes

5. Is the manuscript presented in an intelligible fashion and written in standard English?

Reviewer #1: Yes

Reviewer #2: Yes

6. Review Comments to the Author

Reviewer #1: (No Response)

Reviewer #2: Thank you for addressing my comments. I think the manuscript has been thoroughly, and therefore, has been improved a lot than previous version.

7. PLOS authors have the option to publish the peer review history of their article (what does this mean?). If published, this will include your full peer review and any attached files.

Reviewer #1: No

Reviewer #2: No

---

## [Editor Report · Acceptance letter]

23 Jun 2022

PONE-D-21-05773R1 

Two (or more) for one: Identifying classes of household energy- and water-saving measures to understand the potential for positive spillover 

Dear Dr. Sanguinetti:

I'm pleased to inform you that your manuscript has been deemed suitable for publication in PLOS ONE. Congratulations! Your manuscript is now with our production department. 

Kind regards, 

on behalf of

Dr. Vassilis G. Aschonitis 

Academic Editor

PLOS ONE